# The mitotic chromosome periphery modulates chromosome mechanics

Tania Mendonca [1,2] ✉, Roman Urban[3], Kellie Lucken[1], George Coney[1], Neil M. Kad [3], Manlio Tassieri [4] ✉, Amanda J. Wright [2] ✉ & Daniel G. Booth [1] ✉

In dividing cells, chromosomes are coated in a sheath of proteins and RNA called the mitotic chromosome periphery. This sheath is thought to confer biophysical properties to chromosomes, critical for successful cell division. However, the details of chromosome mechanics, and specifically, if and how the chromosome periphery contributes to them, remain poorly understood. In this study, we present a comprehensive characterisation of single-chromosome mechanics using optical tweezers and an improved broadband microrheology analysis. We extend this analysis to direct measurements of the chromosome periphery by manipulating levels of Ki-67, its chief organiser, and apply a rheological model to isolate its contribution to chromosome mechanics. We report that the chromosome periphery governs dynamic self-reorganisation of chromosomes and acts as a structural constraint, providing force-damping properties. This work provides significant insight into chromosome mechanics and will inform our understanding of the mitotic chromosome periphery's role in cell division.

Chromosomes have intriguing biophysical properties that have been difficult to define despite decades of intense research. Reports on chromosome mechanics vary widely depending on the experimental technique used[1,2]. This variability partly arises from the temporally complex and intrinsically dynamic properties of chromosomes. Depending on the timeframe of the observation window, chromosomes can be described as free polymers diffusing in a viscous nucleoplasm during interphase[3,4] that then transition to a gel-like[5,6] state during mitosis. The complex biophysical properties of chromosomes[7], including emergent behaviours such as condensation, congression and segregation, are influenced by their heterogeneous composition and interactions between the chromatin fibre, proteins, and RNA. While the contribution of some classes of proteins, like the structural maintenance of chromosomes (SMC) protein complexes, to chromosome mechanics has been well studied[8,9], the involvement of other chromatin-interacting biomolecules remains

unclear, or in the case of the mitotic chromosome periphery, is completely unexplored.

The mitotic chromosome periphery (MCP) is a collection of proteins and RNA that redistributes from the disassembled nucleolus to the surface of condensed chromosomes at the onset of mitosis[10–12] and appears to confer biophysical properties to chromosomes, crucial for successful cell division. Chromosomes depleted of the protein Ki-67, the chief organiser of the MCP[13], are lacking in this compartment of over 65 proteins and RNAs[14,15] and become 'sticky' and aggregated[13,16,17]. Additionally, the MCP appears to be multifunctional, with further roles reported in promoting chromosome clustering during late mitosis[18,19], the symmetric distribution of nucleolar material between daughter cells[13], the maintenance of chromosome architecture, either directly[20] or via organisation of its epigenetic landscape[21], and finally as a protector against DNA damage[22]. It has been suggested that some of these functions may be driven by Ki-67 modulating phase separation in

[1]Biodiscovery Institute, School of Medicine, University of Nottingham, Nottingham NG7 2RD, UK. [2]Optics and Photonics Research Group, Faculty of Engineering, University of Nottingham, Nottingham NG7 2RD, UK. [3]School of Biosciences, University of Kent, Canterbury CT2 7NH, UK. [4]Division of Biomedical Engineering, James Watt School of Engineering, Advanced Research Centre, University of Glasgow, Glasgow G11 6EW, UK.
✉e-mail: tania.mendonca@nottingham.ac.uk; manlio.tassieri@glasgow.ac.uk; amanda.wright@nottingham.ac.uk; daniel.booth@nottingham.ac.uk

chromosomes to keep them "individualised", ready for segregation[16,23]. Despite the broad range of critical functions attributed to this chromosome compartment, its biophysical properties have yet to be directly tested. To address this gap, we used optical tweezers to perform direct measurements of the molecular biophysics of the MCP at the single-chromosome level.

In this work, we stretched individual purified chromosomes using optical tweezers to examine their micromechanics. By manipulating Ki-67 expression, we altered the amount of MCP in analysed chromosomes to test the influence of this compartment. Here, conventional force-extension experiments reinforced our understanding that chromosome mechanics are variable, changing with stretching rate. The different stretching rates showed different levels of influence from the MCP, ranging from no effect on bulk elasticity to impacting sequential stiffening at intermediate rates. To extract a comprehensive characterisation of chromosome mechanics and the contribution of the MCP, we have developed a refined new broadband microrheology analysis that can compute mechanical behaviour over seven decades of frequencies from individual chromosomes. This allowed us to analyse not just bulk mechanical properties of single chromosomes at low frequencies but also molecular-scale rearrangements at high frequencies following stretching. Ultimately, our data suggest that the chromosome periphery forms a dynamic external constraint that is independent of the chromosome scaffold. At high frequencies, the MCP dominates chromosome self-reorganisation via 'liquid-like' mechanical behaviour, providing experimental evidence for an often-used description of this compartment.

## Results

### Establishing the single-chromosome approach

Our single-chromosome analysis toolkit includes a stable custom CRISPR-generated Ki-67-mEmerald cell line, enabling direct observation and quantification of Ki-67 on individual human chromosomes. We exploited Ki-67's role as the chief organiser, to modulate MCP chromosome enrichment, by: (1) using Ki-67 specific siRNA to deplete the MCP and (2) transiently transfecting cells with a custom-designed plasmid gRNA vector as part of a CRISPR Activation system, to increase Ki-67 expression, and therefore enhance MCP recruitment (Figs. 1ai–aii and S1, and see Methods for details). Chromosomes with wild-type (WT) or altered MCP load (knockdown 'KD' or overexpressed 'OE') were then isolated from Ki-67-mEmerald cells for single-chromosome analysis using an optical tweezers instrument (C-Trap, LUMICKS) with two independently controlled optical traps, integrated with microfluidics and fluorescence microscopy.

Individual chromosomes were captured between optically trapped pairs of polystyrene bead handles using biotin-streptavidin interactions in a dumbbell configuration (Figs. 1ai, b, S2 and Methods). A stretching force was applied to each chromosome by moving the position of one of the laser beams generating the optical traps and linearly displacing the trapped bead handle. Each optical trap functions as a highly sensitive force transducer for small displacements[24], enabling the measurement of picoNewton forces by tracking the position of the bead handles relative to the respective optical trap centre, i.e., laser beam focus. This picoNewton sensitivity makes optical tweezers ideal for testing chromosome mechanics at

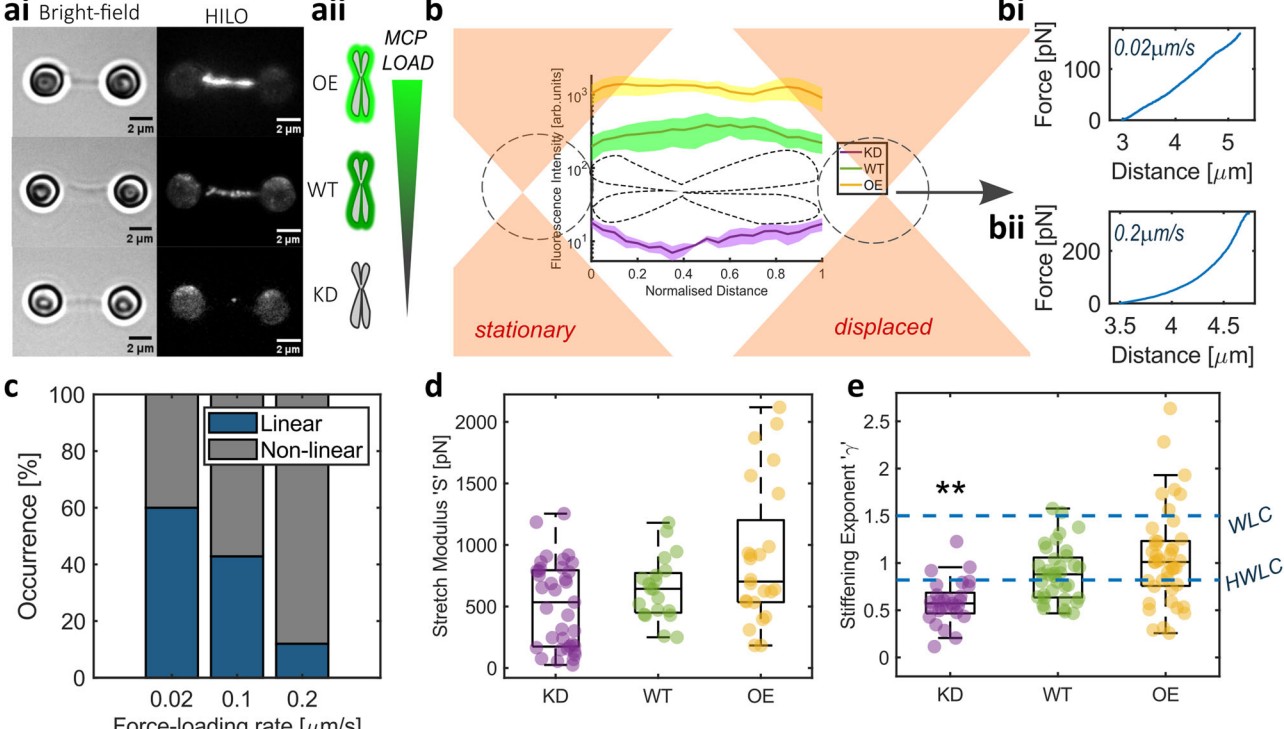

**Fig. 1 | Chromosome mechanics are rate-dependent. ai** Examples of Bright-field and Fluorescence HILO images of Ki-67 over-expression (OE), wild-type (WT) and knock-down (KD) chromosomes in dumbbell configuration. Scale bar = 2 μm. **aii** Schematic of MCP load with Ki-67 expression. **b** Schematic of the force-extension experiments where one optical trap is displaced while the other is kept stationary, to apply stretching forces at a known speed. Fluorescence intensities of chromosomes were analysed to quantify MCP load. Individual examples of force-extension experiments at either 0.02 μm/s (bi) or 0.2 μm/s (bii) rate in WT chromosomes to illustrate the difference in response, showing linear and non-linear behaviour respectively. **c** Occurrence of linear and non-linear mechanical response with different force-loading rates in WT chromosomes. **d** Stretch modulus 'S' was acquired from chromosomes showing linear behaviour at force-extension of 0.02 μm/s (WT n = 20 chromosomes, KD n = 37 chromosomes and OE n = 24 chromosomes). **e** Stiffening exponent γ from chromosomes showing non-linear behaviour at force-extension of 0.2 μm/s (WT n = 36 chromosomes, KD n = 29 chromosomes and OE n = 43 chromosomes) compared to γ values for the worm-like chain (WLC) and hierarchical worm-like chain (HWLC) models. Comparisons to WT (Kruskal–Wallis test), p = 0.001. **d, e** Box plots; Centre: Median, Box bounds: 25th to 75th percentile, Whiskers: minimum and maximum data points (excluding outliers). Data are provided in a Source Data file.

biologically relevant forces. Studies across species and cell types have reported forces ranging from tens to hundreds of picoNewtons acting on individual chromosomes throughout mitotic progression; from congression to bio-orientation and segregation[25–27]. During anaphase A, pulling forces of 200–700 pN have been recorded at chromosomes to move them towards spindle poles in grasshopper spermatocytes[28]. The chromosome manipulations in this study were within this range, with a maximum stretching force of ~400 pN.

### Force-extension measurements confirm that chromosome mechanics are rate-dependent

Chromosomes respond differently to different rates and levels of deformation[1,5,29,30] with predominantly linear spring-like behaviour when stretched slowly (<0.1 μm/s) or non-linear behaviour in most cases when probed at faster rates (Fig. 1bi, bii, c). At a force loading-rate of 0.02 μm/s, which closely corresponds to the recorded speed for oscillations of human chromosomes at the metaphase plate[31], and the rate of directed movement of chromosomes during anaphase in mammalian cells[32] and grasshopper spermatocytes[33], a linear increase in force is required to stretch the chromosome greater distances[2]. The slope of this linear relationship (e.g. Fig. 1bi) defines the stretch modulus 'S', which is the material's spring constant and is also referred to as the doubling force[8,34]. This parameter is analogous to Young's modulus but does not assume material homogeneity. $S$ remains unchanged with varying MCP levels (Fig. 1d). Since $S$ is directly related to the bending stiffness and persistence length of flexible chain polymers, which chromosomes can be modelled as[35], our results suggest that the MCP does not interact with the chromosome scaffold, its backbone.

At a faster stretching rate of 0.2 μm/s relevant to rates at which microtubule depolymerisation-led chromosome motion was observed in vitro[36], chromosome moving chromo-kinesin motor proteins function in mammalian cells[37] and SMC motors have been recorded extruding DNA loops in yeast[38], the force-extension relationship bears a non-linear form in a majority of tested chromosomes (Fig. 1bii, c). Such non-linear stiffening can be attributed to the chromosome network microstructure and force transmission through cross-links[39]. This stiffening 'κ' bears a power-law relationship with force 'F', where $\kappa = aF^{\gamma}$[40]. In WT chromosomes, the exponent 'γ' was 0.88 [0.78, 0.98] (mean ± 95% confidence intervals), which is in agreement with literature[40] and lower than the γ value of 3/2 attributed to the worm-like chain model used to describe the mechanics of double-stranded DNA[40–42]. This decreased stiffening rate has been explained using a Hierarchical Worm-Like Chain (HWLC) model[40], which postulates chromosomes to be hierarchical assemblies of elements with distinct mechanical properties, each acting as a flexible worm-like chain. The γ value drops to 0.56 [0.46, 0.66] with the loss of the MCP (Fig. 1e; Kruskal–Wallis test with multiple comparisons, WT vs KD $p = 0.001$). This lower γ value suggests a disruption of cross-linked structural elements that contribute to force transmission and the sequential stiffening response with increasing force[39]. However, γ did not significantly change for OE chromosomes (1 [0.9, 1.2]; Kruskal–Wallis test with multiple comparisons, WT vs OE $p = 0.43$), indicating no discernible change to the hierarchical organisation of the chromosome in this case. The structural organisation of the MCP remains underexplored. Current speculation places Ki-67 in a polarised brush conformation on the chromosome surface, acting as a scaffold for other MCP proteins and RNA[16,43,44]. This structural model predicts an increase in spatial density of the MCP in OE chromosomes, which may result in a disordered MCP cross-linked network, explaining the broad variation in our results. Individual chromosomes from all treatments exhibited consistent mechanical behaviour with repeated extensions for forces of up to 150 pN, as with previously reported findings[29,40].

The HWLC model assumes that chromosomes exhibit a predominantly elastic response to stress[40], which contrasts with prior reports of viscous relaxation in chromosomes at force-extension rates of 100 μm/s[30]. These results, when considered in isolation, highlight the limitations of discrete single-frequency measurements in accurately capturing the dynamic, time-dependent mechanical properties of chromosomes and the complex contributions of their associated structures, such as the MCP.

### Broadband microrheology reveals self-organisation dynamics in single chromosomes

This prompted us to develop a broadband microrheology approach for single chromosomes in a single step, as opposed to performing a multitude of single-frequency stretching experiments with different rates and extents. While micromanipulation tools such as optical tweezers and magnetic tweezers have been used for microrheology studies of interphase chromosomes[3,45,46], their potential for broadband characterisation of mitotic chromosome dynamics remains underexplored. In whole cells[47] and tissue cultures[48], optical tweezers have been used for passive broadband microrheology, where the optical trap is used purely for the spatial confinement of the bead probe. Active-passive approaches that combine optically driven perturbation of the bead probe with broadband passive measurements[49,50] have been developed for microrheology of viscoelastic liquids. However, these active-passive approaches are multi-step procedures that require sophisticated frequency and phase modulation for the active perturbation. In contrast, we introduce a bespoke single-step optical tweezer-based stretch-hold method, inspired by a similar approach used for microrheology of cells with atomic force microscopy[51,52], to extract the mechanics of individual mitotic chromosomes across a broad frequency range. The complex stiffness $\kappa^*(\omega)$ derived from this method provides information about the chromosome's viscoelastic properties. It is defined as the ratio of the Fourier transforms of force $F(t)$, measured as the picoNewton force acting on the chromosome with time, and deformation $\lambda(t)$, the extension of the chromosome in nanometres over time. $\kappa^*(\omega)$ is a complex number with real and imaginary parts that describe the elastic $\kappa'(\omega)$ and viscous $\kappa''(\omega)$ components of the chromosome mechanical response.

Using the chromosome dumbbell configuration, chromosomes were stretched by displacing the position of one of the two optical traps a fixed distance of 1 μm, at a rate of 100 μm/s (Fig. 2a–b). Although chromosomes would not be deformed at 100 μm/s in nature, these fast manipulations allow us to probe molecular-scale interactions along with macro-scale mechanical properties to understand the contribution of the MCP. This fast stretching was followed by a dwell period of 2 min, during which the positions of both optical traps were held constant. During the dwell period, the chromosome continued to extend in the stretched state, shifting the relative position of the bead handles with respect to the centres of the optical traps which resulted in a change in force acting on the bead handles (Fig. 2c). Force acquisition at high frequency (2.5 MHz) was extracted during the dwell period (Fig. 2d) and provided continuous broadband viscoelastic data over seven decades of frequencies ($10^{-2}$–$10^5$ rad/s) from single chromosomes. This approach far surpasses oscillatory microrheology methods[40,53], which derive mechanical properties at discrete frequencies and cannot extract the full frequency spectrum from individual chromosomes. Our data reveal fluid-like behaviour at high frequencies ($10^1$–$10^3$ rad/s) following deformation in all tested WT chromosomes (Fig. 2e). This fluid-like response likely reflects energy lost to the system from internal friction, probably as a result of the reorganisation of the chromosome network. At lower frequencies, after a characteristic time of approximately 100 ms, the chromosomes equilibrate to a gel-like state dominated by elasticity, with minimal changes in viscoelasticity observed across nearly three decades of frequency.

Measurements were also recorded at a slower force-loading rate of 0.2 μm/s to investigate the influence of loading rate on

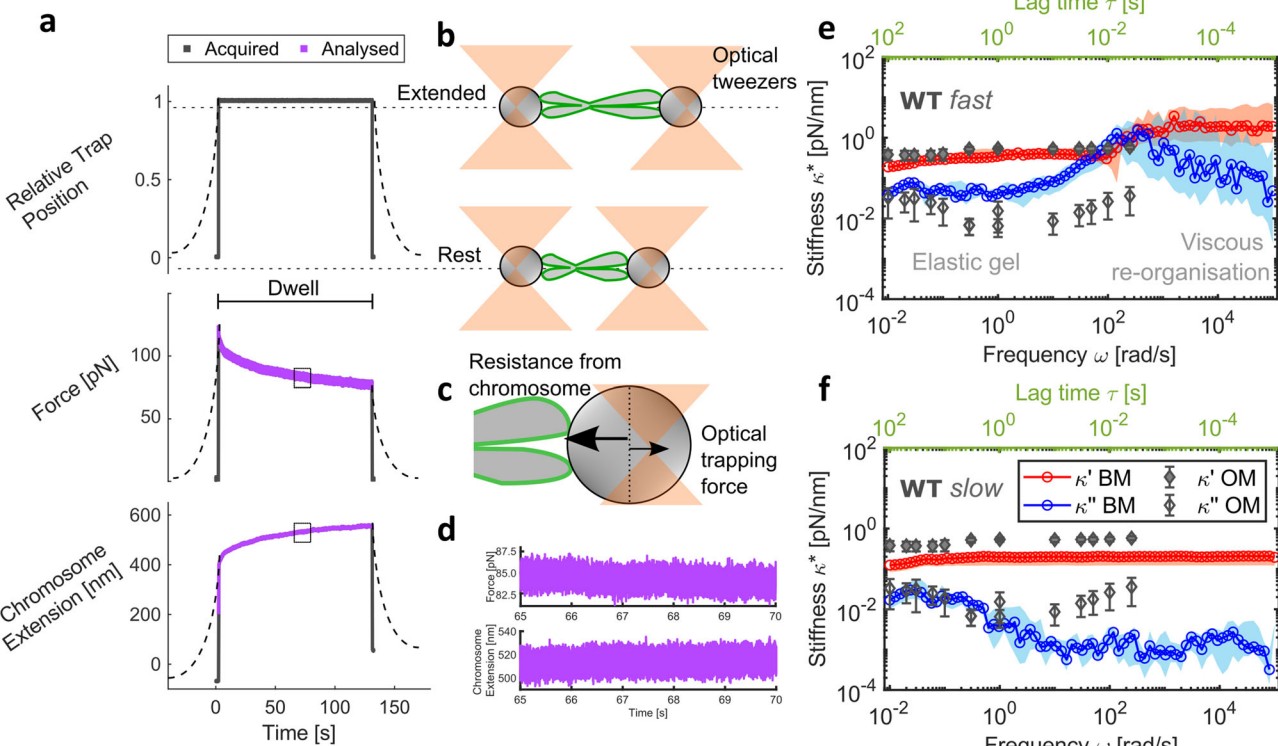

**Fig. 2 | Broadband microrheology of chromosomes. a** Schematic of micro-rheology experimental procedure. Dashed lines represent data at a force-loading rate of 0.2 µm/s and solid lines for 100 µm/s. Data in purple were analysed to provide broadband mechanical response. **b** Schematic representation of the tweezer and chromosome positions from (**a**) **c** Opposing forces experienced at bead handles (one shown) in the non-equilibrium state. **d** Zoomed-in sub-region of the analysed force at one bead and chromosome extension data. **e** Complex stiff-ness $\kappa^*(\omega)$ with frequency (bottom axis in black) and lag time $\tau$ (top axis in green)

from broadband microrheology (BM) of WT chromosomes at 100 µm/s (median and 95% CI; $n = 14$ chromosomes), highlighting regions of viscous reorganisation and gel-like behaviour. Data in blue are the viscous modulus $\kappa''(\omega)$ and in red are the elastic modulus $\kappa'(\omega)$. **f** $\kappa^*(\omega)$ at 0.2 µm/s force-loading rate (median and 95% CI; $n = 15$ chromosomes) of WT chromosomes. **e** and **f** are both overlaid with oscilla-tory microrheology (OM) data from Meijering et al. (2022)[40],[74]. Schematics shown are not to scale. Data are provided in a Source Data file.

microrheology in WT chromosomes (Fig. 2f). At this reduced loading rate, the reorganisation within the chromosome network occurred faster than the applied strain deformation, resulting in the absence of detectable self-reorganisation dynamics at short timescales. Instead, the measurements revealed a predominantly gel-like response throughout the entire duration of the experiment. Notably, our com-plex stiffness $\kappa^*(\omega)$ values agree well with previously reported oscilla-tory microrheology measurements of chromosomes[40] over four decades of frequency (Fig. 2e, f). At intermediate frequencies (~100 rad/s), oscillatory measurements show a similar trend in the viscous modulus $\kappa''(\omega)$ to the 100 µm/s measurements. This suggests that the brief cross-over between the $\kappa''(\omega)$ and $\kappa'(\omega)$, where viscous self-reorganisation processes dominate, might have been captured if oscillatory measurements were continued at higher frequencies than those reported[40] (Fig. 2f). These results highlight the need for applying fast force-loading rates and a broadband approach to effectively cap-ture the full spectrum of chromosome molecular biophysics, enabling the study of chromosome periphery mechanics in previously unat-tainable detail.

### Chromosome viscoelastic response is modulated by the chro-mosome periphery

Altering chromosome periphery levels changes the chromosome vis-coelastic response recorded using our broadband microrheology method with a fast force-loading rate of 100 µm/s (Fig. 3ab and Fig. 2e). The ratio of viscous $\kappa''(\omega)$ to elastic $\kappa'(\omega)$ components of the chro-mosome's mechanical response defines the loss tangent $\tan\delta$, which

provides insight into the force damping or energy dissipation potential of the chromosome. Regardless of chromosome periphery status, $\tan\delta$ values for all chromosomes peak at relatively high frequencies ($10^2$–$10^3$ rad/s), where energy dissipation is at maximum due to molecular reorganisation, before reaching a minimum where the chromosome equilibrates to an elastic gel state (Fig. 3c). WT chro-mosomes exhibit a $\tan\delta$ peak value of 2.7 [1.2, 5.4; 95% CI]. KD chro-mosomes show a distinct absence of crossover between $\kappa''(\omega)$ and $\kappa'(\omega)$, and consequently, their $\tan\delta$ values remain below 1, peaking at 0.6 [0.4, 0.7; 95% CI], indicating a consistent predominantly elastic material with low energy dissipation potential. Higher force damping in the presence of the MCP provides a possible mechanistic explana-tion for its damage mitigation properties noted previously[22]. In con-trast, OE chromosomes varied in their response, with some exhibiting behaviour similar to WT chromosomes, while others displayed more KD-like behaviour, with overall $\tan\delta$ peak values of 0.5 [0.4, 1.6; 95% CI] (Fig. 3d). The characteristic time (defined as the inverse of the characteristic frequency) when $\tan\delta$ values peak differs significantly between WT, KD, and OE chromosomes. Most WT chromosomes reach their peak at 10 ms (±1 ms) post-deformation, while KD chromosomes peak earlier at 4 ms (±1 ms). OE chromosomes alternate between these two peak times, with 60% peaking at 4 ms. Additionally, all WT and OE $\tan\delta$ data also show a smaller peak at 4 ms (where not dominant) (Fig. 3c), suggesting the presence of two distinct molecular mechan-isms driving chromosome relaxation. Consistent with this, the $\tan\delta$ minima for the different conditions were also divergent, with WT chromosomes exhibiting a minimum at 1.5 s (±1 s), KD chromosomes

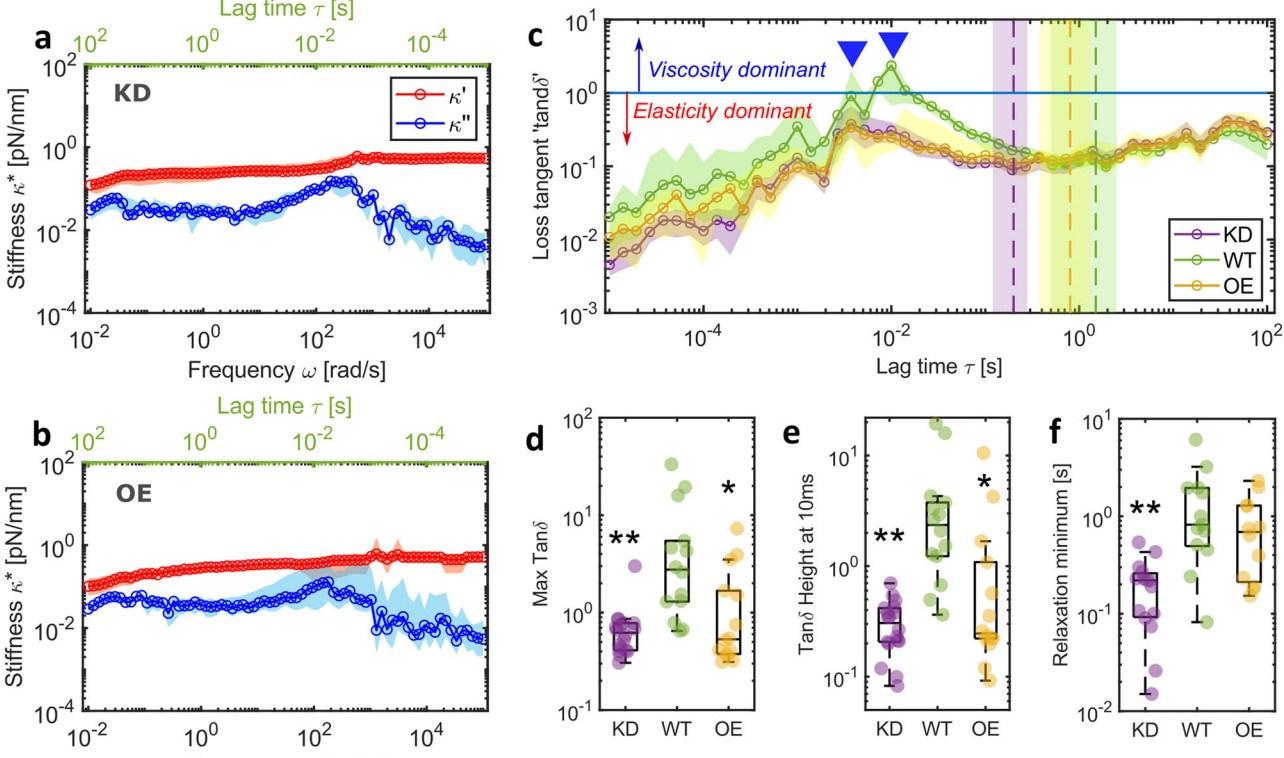

**Fig. 3 | Chromosomes show distinct relaxation dynamics in the presence and absence of the MCP. a** Complex stiffness $\kappa^*(\omega)$ of KD chromosomes (median and 95% CI; $n = 16$ chromosomes) and **b** OE chromosomes (median and 95% CI; $n = 15$ chromosomes) compared to WT chromosomes in Fig. 2e. Data in blue are the viscous modulus $\kappa''(\omega)$ and in red are the elastic modulus $\kappa'(\omega)$. **c** tan $\delta$ values for the three conditions (mean ± 95% CI). Blue line represents tan $\delta$ value of 1 and values higher than this indicate viscosity is greater than elasticity. Inverted triangles highlight peak reorganisation times for WT and both KD and OE chromosomes. Dashed vertical lines with coloured regions show mean ± 95% CI of time at relaxation minimum. **d** Maximum value of tan $\delta$ for individual chromosomes. Comparisons to WT (Kruskal−Wallis test): KD $p = 0.001$, OE $p = 0.008$. **e** tan $\delta$ value at 10 ms for individual chromosomes. Comparisons to WT (Kruskal−Wallis test): KD $p = 0.0002$, OE $p = 0.002$. **f** tan $\delta$ minimum after relaxation highlighted by the vertical dashed lines in c, comparisons to WT (Kruskal−Wallis test): KD $p = 0.0004$, OE $p = 0.618$. **c**-**f** WT $n = 14$ chromosomes, KD $n = 16$ chromosomes, OE $n = 14$ chromosomes. Box plots: Centre: Median, Box bounds: 25th to 75th percentile, Whiskers: minimum and maximum data points (excluding outliers). Significance values: $*p < 0.05$, $**p < 0.001$, $***p < 0.0001$. Data are provided in a Source Data file.

relaxing significantly earlier at 0.2 s (±0.08 s), and OE chromosomes displaying an intermediate relaxation minimum at 0.8 s (±0.5 s). These results collectively indicate MCP-dependent reorganisation mechanisms, which are absent in KD chromosomes lacking the MCP. Furthermore, reorganisation appears to be suppressed in OE chromosomes, suggesting that an optimal MCP load is required for normal mechanical behaviour.

The MCP is enriched in proteins with intrinsically disordered domains[14]. Ki-67 and NPM1 are notable examples[23,54] where the disordered domain has been shown to possess patterned charge distributions[54] that enable promiscuous interactions with other proteins and RNA[14,44]. Indeed, unbinding of transient cross-links from electrostatic or van der Waals interactions has been shown to produce pronounced maxima in viscous dissipation at frequencies of $10^{-2}$–$10^2$ Hz in actin networks[55] and could explain the MCP dependent relaxation dynamics seen in our data. Furthermore, in Ki-67 OE chromosomes with an enriched (Fig. S1) and therefore more densely packed MCP if we assume the brush model[16], an increase in charged domains could alter their viscoelastic properties through electrostatic or steric repulsion to reduce molecular flexibility, resulting in gel-like instead of fluid-like behaviour[56]. MCP independent reorganisation, on the other hand, seen at earlier characteristic times and which appears to be the main relaxation mechanism in KD chromosomes, has been linked to SMC activity[6]. The SMC arms in the condensin holocomplex have been reported to be flexible and able to change conformation in

millisecond time scales[57], in good agreement with our data (Fig. 3c). Other structural proteins such as TopIIα and Kif4A are also potential candidates involved in this reorganisation and will need to be tested in future studies. The stretch-hold microrheology method introduced here would enable such investigations.

## The Burgers model attributes chromosome fluidity to the chromosome periphery

Finally, we sought to fit a rheological model to parameterise our broadband experimental data and further isolate the contribution of the MCP to chromosome mechanical behaviour. The Burgers model, previously used to describe the linear viscoelastic properties of nuclei in intact cells[58] and multicellular spheroids[59], was used to interpret chromosome dynamics for the first time (Figs. 4a and S4). This four-element model consists of two elastic $\kappa_1$ and $\kappa_2$ and two viscous damping terms $\eta_1$ and $\eta_2$ (Fig. 4b). $\kappa_1$ and $\kappa_2$ describe the plateaus in elastic modulus $\kappa'(\omega)$ at low and high frequencies, respectively, while $\eta_1$ in combination with $\kappa_1$ characterises the transition in viscoelasticity at intermediate frequencies, and $\eta_2$ corresponds to the viscosity of the system at long timescales (i.e., low frequencies).

Our model fitting indicates that KD chromosomes exhibit reduced $\kappa_2$ (Kruskal−Wallis test with multiple comparisons, WT vs KD $p = 0.001$) and $\eta_2$ values (Kruskal−Wallis test with multiple comparisons, WT vs KD $p < 0.0001$) compared to WT chromosomes, while showing similarities to OE chromosomes.

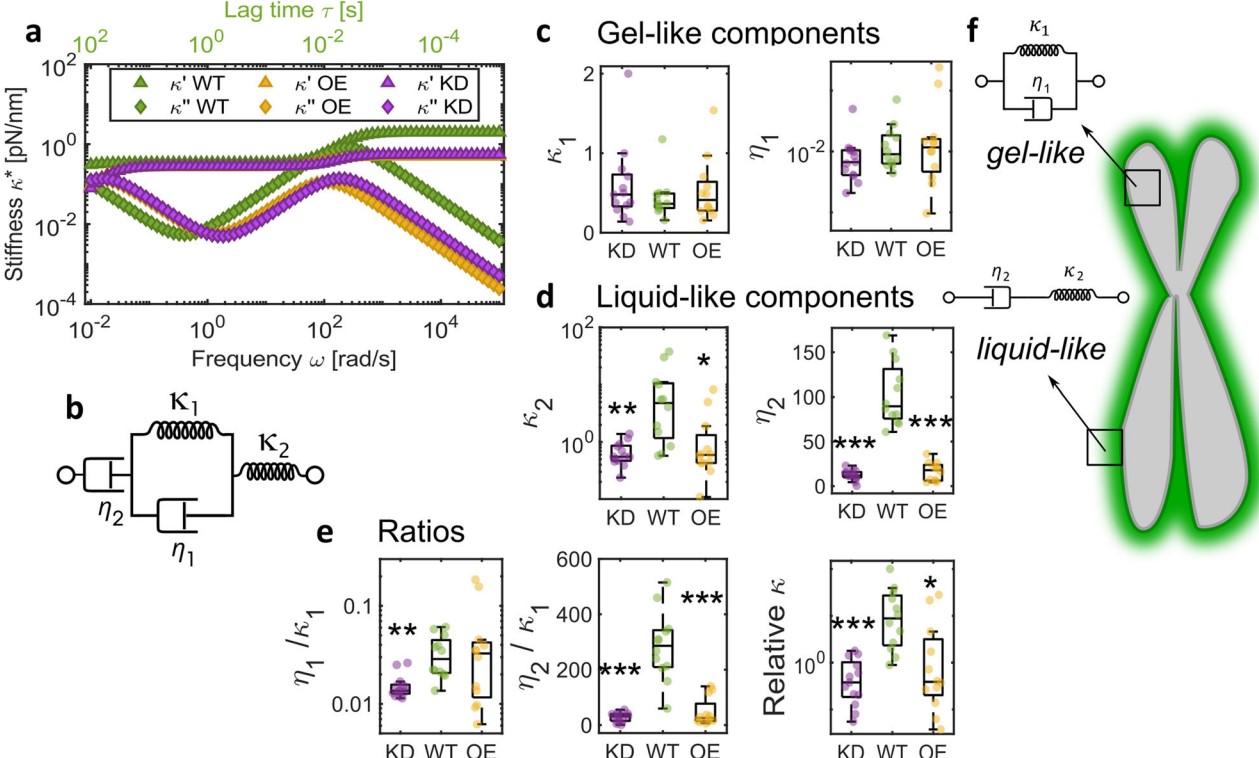

**Fig. 4 | Chromosome mechanical behaviour explained using the Burgers model. a** Burgers model fits to average WT ($n = 14$ chromosomes), KD ($n = 16$ chromosomes) and OE data ($n = 15$ chromosomes). **b** Spring-dashpot schematic of the Kelvin representation of Burgers Model. **c, d** Comparisons of parameters extracted by fitting experimental data to the Burgers model individually $\kappa_1$ comparisons to WT (Kruskal–Wallis test): KD $p = 0.57$, OE $p = 0.99$; $\eta_1$ comparisons to WT (Kruskal–Wallis test): KD $p = 0.23$, OE $p = 0.94$; $\kappa_2$ comparisons to WT (Kruskal–Wallis test): KD $p = 0.001$, OE $p = 0.015$; $\eta_2$ comparisons to WT (Kruskal–Wallis test): KD $p < 0.0001$, OE $p < 0.0001$. **e** Ratios of fit parameters; $\eta_1/\kappa_1$

comparisons to WT (Kruskal–Wallis test): KD $p = 0.001$, OE $p = 0.77$; $\eta_2/\kappa_1$ comparisons to WT (Kruskal–Wallis test): KD $p < 0.0001$, OE $p < 0.0001$; relative $\kappa$ comparisons to WT (Kruskal–Wallis test): KD $p = 0.0001$, OE $p = 0.022$. For **c–e**, WT $n = 14$ chromosomes, KD $n = 16$ chromosomes, OE $n = 15$ chromosomes. Box plots: Centre: Median, Box bounds: 25th to 75th percentile, Whiskers: minimum and maximum data points (excluding outliers). Significance values: *$p < 0.05$, **$p < 0.001$, ***$p < 0.0001$. **f** Schematic of a chromosome to show gel-like properties are associated with the whole chromosome, while the MCP shows liquid-like dynamics. Data are provided in a Source Data file.

Furthermore, while $\kappa_1$ and $\eta_1$ remain unchanged regardless of MCP status, the ratio between these two parameters is significantly shifted in KD chromosomes (Kruskal–Wallis test with multiple comparisons, WT vs KD $p = 0.001$), pointing to differences in relaxation processes at intermediate frequencies, probably associated with conformational changes and transient cross-linking, reflecting our $\tan\delta$ results (Fig. 3c). Similarly, the ratio between $\eta_2$ and $\kappa_1$, relating to viscoelastic behaviour at low frequencies likely driven by large-scale protein interactions, shows significant change in KD chromosomes (Kruskal–Wallis test with multiple comparisons, WT vs KD $p < 0.0001$) but also in OE chromosomes (Kruskal-Wallis test with multiple comparisons, WT vs KD $p < 0.0001$) which may be due to disordered interactions in a presumably overcrowded OE MCP. The elastic parameters extracted from the model reveal a smaller relative change in stiffness in KD and OE chromosomes in comparison to WT chromosomes (Fig. 4e; Kruskal–Wallis with multiple comparisons of relative $\kappa$ from model fitting, WT vs KD $p = 0.0001$, WT vs OE $p = 0.022$), but no statistically significant differences with MCP status in elasticity at low frequencies (Fig. 4c, in agreement with our single frequency force–extension results 1d). The Burgers model can be interpreted as a combination of a Kelvin–Voigt solid, expressed by $\kappa_1$ and $\eta_1$, likely reflecting the contribution of the chromosome as a whole, and a Maxwell liquid, represented by $\kappa_2$ (instantaneous response) and $\eta_2$ (long time viscous flow), which may capture the contribution of the MCP among other

chromosomal elements (Fig. 4f). Notably, Ki-67 and the MCP have often been described as exhibiting 'liquid-like' properties[18,23,54] and our data represents direct measurements of these properties.

The chromosome periphery has been shown to be self-supporting, retaining its structure even after DNA and RNA digestion[60]. The precise nature of linkages that maintain the MCP compartment remains unclear; however, it has been shown to function independently of TopoII$\alpha$ and condensins[20], which corroborates our single frequency stretch modulus results. Transient cross-links driven by charged interactions with intrinsically disordered proteins like Ki-67 may explain the dynamic, 'fluid' nature of the MCP. Such charged interactions have recently been shown to influence chromosome fluidity and can be modified by altering the ionic strength of the experimental buffer[53]. Chromosomes behave as polyelectrolyte gels where charged interactions modulate osmotic pressure, as described by Donnan's theory[53,61], to support chromosome structure. These polyelectrolyte properties may be driven by the MCP alongside histones. Our data, however, show that OE chromosomes, which are expected to possess an excess of charged domains, do not appear decondensed (Fig. S3d). It is therefore probable that (i) the experimental polyamine (PA) buffer (see Methods) is able to compensate for this increase in charged domains, or (ii) the MCP may not significantly contribute to chromosome polyelectrolyte properties and therefore, chromosome condensation. It has been previously shown that while removing the MCP by silencing Ki-67 expression

does not prevent chromosome condensation, its disruption at a later stage of mitosis results in distorted chromosomes[20].

Our data provide evidence of the influence of the MCP on stiffening behaviour (Fig. 1e) of individual chromosomes, indicating a periphery-based structural network, independent of the chromosome scaffold. However, a Ki-67-driven structural constraint has previously been dismissed in favour of nucleosome-nucleosome interactions at the chromosome periphery[62]. Nevertheless, these nucleosome interactions do not fully account for our findings, which highlight a distinct MCP-driven chromosome relaxation process, warranting further investigation.

## Conclusion

In summary, this study highlights the importance of broadband analysis for capturing the time-dependent dynamics of chromosomes. Our data reveal a timeline of dynamic mechanical events following chromosome stretching, spanning timescales from tens of microseconds to minutes, and elucidate the direct role of the MCP in regulating chromosome fluidity and structural integrity. Increased Ki-67 expression is a hallmark of cancer progression, and there is strong interest in using this protein as a therapeutic target[63–65], paving a translational path for our findings. Furthermore, temporal mapping of mechanical responses provides a foundation for exploring the role of charged interactions in chromosome dispersal and volume phase separation, and more broadly, the biophysics of phase separation, another emerging therapeutic target[66]. Finally, broadband microrheology of single chromosomes opens avenues for investigating the contributions of other chromosome-associated structures, many of which are still poorly defined.

## Methods

### Generation of a HeLa Ki-67-mEmerald cell line

The HeLa parental cell line was sourced from DSMZ. DNA oligonucleotides (Forward 5′ ACATGGACATGAGCCCCCTG 3′, Reverse 5′ GATAGTTCTGGGGCCTCAGG 3′) used for gRNA synthesis were designed using the Benchling CRISPR gRNA Design Tool available at (www.benchling.com, accessed on 15 May 2022) and ordered through Sigma Aldrich. gRNA was annealed together and cloned into the Bbs1 restriction site of the pX330-U6-Chimeric_BB-CBh-hSpCAas9-hGem (1/110) vector (Addgene #71707) before transformation into NEBR 5-alpha Competent *E. coli* (New England Biolabs, Ipswich, Massachusetts, USA). DNA was prepared from single colonies before sequence verification using an Applied Biosystems™ (Waltham, Massachusetts, USA) 3130xl genetic analyser. Homology-directed repair (HDR) plasmids designed to introduce mEmerald to the C-terminus of *MKI67* were synthesised using a pUC18 backbone and ordered from GenScript (Piscataway, NJ, USA). The plasmids also included a puromycin resistance cassette to facilitate efficient positive selection of transfected cells, as well as a T2A sequence to separate the puromycin-resistant protein from the mEmerald-fused Ki-67 protein. HeLa cells were nucleofected with the HDR plasmid and the gRNA cas9 plasmid and allowed to recover for 3 days before selection with G418 for 7 days. Cells were then treated with puromycin (Fisher Scientific) for 3 days and single-cell sorted into ninety-six-well plates. Clones were screened by PCR genotyping.

### Overexpression construct

A gRNA sequence (Forward 5′ CACCGATTTGACAGAAAAATCGAACC AAACT 3′, Reverse 5′ AAACAGTTTGGTTCGATTTTTCTGTCAAATC 3′) was cloned into a CRISPRa plasmid (Addgene #175572) containing an inactivated CRISPR/Cas9 fusion protein with the VPR transcriptional activation domain, and mCherry as a fluorescent reporter.

### Tissue culture and transfection

The Ki-67-mEmerald cell line was maintained in DMEM supplemented with 10% foetal bovine serum (FBS) and 1% antibiotics (penicillin-streptomycin) at 37 °C in 5% $CO_2$ and was regularly tested and confirmed free of mycoplasma contamination. To knock down Ki-67, the Ki-67-mEmerald cell line was transiently transfected with 0.8 μg siRNA (Ki-5 in ref. 13) per mL of growth media using Lipofectamine RNAiMax (Invitrogen) according to the manufacturer's instructions and analysed after 72 h. Ki-67 was transiently overexpressed in the Ki-67-mEmerald cell line with 0.8 μg of the overexpression construct per mL of growth media using jetPRIME (VWR, Lutterworth, Leicestershire, UK) transfection reagent (PolyPlus).

### Chromosome isolation

Ki-67-mEmerald cells were grown to the exponential growth phase in T125 flasks before transfection for 72 h to knock out or overexpress Ki-67. Growth media was supplemented with biotin (50 mM) 24 h before chromosome isolation using a protocol similar to previously described[40,67] using polyamine (PA) buffer to enhance purity of isolated chromosomes and retain chromosome structure[67–69]. Briefly, cells were arrested in prometaphase with Nocadazole (100 ng/mL) treatment for 14 hs. Arrested cells were collected by mitotic shake-off, centrifuged at 300g for 5 min and incubated in a swelling buffer (10 mL of 75 mM KCl and 5 mM Tris-HCl, pH 8.0 for $10^7$ cells) for 30 min at 37 °C. The swollen cells were then centrifuged at 4 °C and re-suspended in 8 mL of cold PA buffer (15 mM Tris-HCl (pH 8.0), 2 mM EDTA, 0.5 mM EGTA, 80 mM KCl, 20 mM NaCl, 0.5 mM spermidine, 0.2 mM spermine and 0.2% Tween-20) before mechanically breaking in the presence of protease inhibitors (Pierce) and phosphatase inhibitors (PhosSTOP, Roche) in a 15 mL dounce homogeniser (Kimble) with 40 strokes of a tight pestle on ice. The homogenised suspension was cleared of debris three times before glycerol gradient fractionation (2 mL each of 60% and 30% glycerol in PA) by centrifugation at 1750g for 30 min at 4 °C. Isolated chromosomes were collected from the 60% glycerol fraction. Chromosomes were stored in 60% glycerol at −20 °C and used within three months of isolation.

### Chromosome micromanipulations

A C-trap Edge-450 (LUMICKS) instrument at the University of Nottingham and another at the University of Kent were used to analyse chromosome dynamics. These instruments combine optical tweezers (up to 4 optical traps) and microfluidics with multichannel laminar flow, which allows for the incremental assembly of the experimental unit (Fig. S2). The instrument was used in the dual-trap mode with a 50% split of laser power between the two optical traps. A pair of streptavidin-coated polystyrene beads (3 μm diameter, Bangs Laboratories) in PA buffer was optically trapped in the first microfluidic channel with the two optical traps. Trap stiffness was calibrated for every new bead pair in the absence of fluid flow and was 0.5 ± 0.03 pN/nm at each bead. The trapped bead pair was then moved across to the channel flowing chromosomes suspended in PA buffer (channel 2) to capture a single biotinylated chromosome first by binding to one bead. The fluid flow oriented the attached chromosome such that a free end was available for the second bead to be brought into contact with it to form a dumbbell unit.

Chromosome attachment was biased along the telomeric ends due to the orientation of the chromosomes within the fluid flow. Non-biotinylated chromosomes did not attach to the beads. Sister chromatids are conjoined in nocodazole-synchronised chromosomes due to cohesins not yet being degraded in prometaphase. These isolated chromosomes are thread-like, with each chromatid being ≃0.6μm across and much smaller than the diameter of the beads (3 μm), which ensured contact between both sister chromatids at each bead in the telomeric conformation. Experimental manipulations were performed using this dumbbell configuration with the optical traps aligned along

the $x$-axis of the C-trap imaging system. At the start of every experiment, before any stretching forces were applied to the chromosome, the forces at both beads in the chromosome dumbbell were zeroed to set a baseline force reference. One optical trap was then stepped a small distance to ensure attachment of the chromosome to both beads and to bring it to its natural length $L_0$, defined as the distance between the bead handles before any further stepping of a bead resulted in resistive force from the chromosome above the set baseline. This step rotated one or both beads within their respective optical trap in most cases to bring the chromosome to $L_0$ and in alignment with the $x$-axis of the imaging system and the dumbbell, before any deformation of the chromosome could occur (Fig. S3). Further manipulations were performed in a third microfluidic channel containing only PA buffer, as detailed in the sections below.

Fluorescence images were simultaneously acquired during experiments using HILO (pseudo TIRF) microscopy on the C-trap. Intensities of individual chromosomes (see Image analysis section) were used to exclude chromosomes from untransfected cells within each treatment group. Only chromosomes between 2 and 5 μm in length were analysed to ensure single chromosomes with telomeric attachment were being tested (Fig. S3d). Damaged and unravelled chromosomes were discarded on visual inspection. Analysis was performed in MATLAB 2022b (MathWorks) using custom scripts.

### Force–extension experiments

Single frequency force-extension experiments (Fig. 1) involved keeping one optical trap in the dumbbell unit stationary while linearly displacing the second trap position along the $x$-axis at a fixed rate of 0.02 μm/s to measure linear elasticity and 0.2 μm/s to examine chromosome non-linear stiffening behaviour using the inbuilt force spectroscopy module in Bluelake version 2.5.1 (LUMICKS) which removed subjectivity during chromosome manipulation. Chromosome response to stretching is variable (Fig. 1c), so only chromosomes showing linear behaviour at 0.02 μm/s (WT $n = 20$ chromosomes, KD $n = 37$ chromosomes and OE $n = 24$ chromosomes) and non-linear behaviour at 0.2 μm/s (WT $n = 36$ chromosomes, KD $n = 29$ chromosomes and OE $n = 43$ chromosomes) were analysed. In each case, the optical trap was continuously displaced until the resistance from the chromosome was greater than that which could be overcome by the optical trapping force, resulting in one or both beads leaving the optical trap.

The stretch modulus $S$ was computed as the slope of force against normalised chromosome extension, $(L(t) - L_0)/L_0$, where $L(t)$ is the length of the chromosome at the given time and $L_0$ is the natural length of the chromosome. Non-linear stiffening analysis was performed as described previously[40]. In brief, data were smoothed using a moving average function with a window size of 1/15 of the data points in the measurement. Stiffening exponent $\gamma$ was computed by fitting an exponential function of the form

$$\kappa = aF^\gamma \tag{1}$$

to the numerical gradient $\kappa$ of the force-extension curve against force $F$.

### Microrheology

A new broadband microrheology technique was developed to analyse the time-dependent response of single chromosomes to a stretching force (Fig. 2). At the start of the experiment, the chromosome was fully extended to its natural length $L_0$ in the chromosome dumbbell configuration such that any further extension by moving one of the bead handles registered an increase in force acting on each bead. Force was exerted on the captured chromosome by laterally displacing the position of one of the two optical traps by 1 μm at speeds of 100 μm/s (WT $n = 14$ chromosomes, KD $n = 16$ chromosomes, OE $n = 15$

chromosomes). To examine the effect of force-loading rates on chromosome response, force-loading at 0.2 μm/s (only WT $n = 15$ chromosomes) was also performed. In these slower force-loading rate measurements, the moving optical trap was displaced until a force readout of 150 pN on the bead was detected. The chromosome was then held in the extended position for around 2 min before restoring it to its original state by returning the displaced optical trap to the start position. High frequency (2.5 MHz) force data over the 2 min dwell period was used in the microrheology analysis. Data collection was automated using the force distance module in Bluelake version 2.5.1 (LUMICKS). The force acting at each optically trapped bead can be calculated as

$$\vec{F} = -\kappa_{OT}\vec{x} \tag{2}$$

where $\vec{x}$ is the shift in the bead's position relative to the optical trap centre and $\kappa_{OT}$ is the stiffness of the optical trap.

The force exerted by the displaced bead on the chromosome is transmitted along the chromosome to the stationary bead, which reacts with equal magnitude and opposite direction to hold the chromosome in place. The stretching force $F(t)$ acting on the chromosome was therefore calculated as the average of the force measured at both bead handles for accuracy. The deformation $\lambda(t)$ is calculated as the extension of the chromosome in nanometres:

$$\lambda(t) = L(t) - L_0 \tag{3}$$

The high-frequency position information for each optically trapped bead was computed by subtracting $\vec{x}$ from the position of the optical trap holding that bead. The broadband mechanical response of individual chromosomes was computed as complex stiffness

$$\kappa^*(\omega) = \frac{\hat{F}(\omega)}{\hat{\lambda}(\omega)} = \kappa'(\omega) + i\kappa''(\omega) \tag{4}$$

$\hat{F}(\omega)$ and $\hat{\lambda}(\omega)$ are the Fourier transforms of $F(t)$ and $\lambda(t)$, respectively.

The loss tangent $\tan\delta$ (Fig. 3c) was calculated as,

$$\tan\delta = \frac{\kappa''(\omega)}{\kappa'(\omega)} \tag{5}$$

$\kappa^*(\omega)$ and $\tan\delta$ were computed using a MATLAB application i-Rheo C-Stretch[70] (see Code availability and Fig. S5) that implements a Fourier transform method introduced previously[52,71,72]. Like any numerical method, this analytical approach is also influenced by the level of noise present in the input measurements. The impact of noise is most pronounced on the smaller of the two viscoelastic components, as the signal-to-noise ratio here will be lower. In this study, the elastic modulus dominates across most of the explored frequency range, particularly at high frequencies, making it less susceptible to measurement noise. In contrast, the viscous component, being predominantly smaller, is more sensitive to noise-induced fluctuations.

The characteristic minimum time in $\tan\delta$ was extracted as the time at which an exponential fit to the data from each individual chromosome reached the minimum value.

### Model fitting

The Kelvin representation of the Burgers model can be expressed as,

$$F(t) + \left(\frac{\eta_1}{\kappa_1} + \frac{\eta_2}{\kappa_1} + \frac{\eta_2}{\kappa_2}\right)\dot{F}(t) + \frac{\eta_1\eta_2}{\kappa_1\kappa_2}\ddot{F}(t) = \eta_2\dot{\lambda}(t) + \frac{\eta_1}{\kappa_1}\eta_2\ddot{\lambda}(t) \tag{6}$$

Here, $\kappa_1$, $\kappa_2$, $\eta_1$ and $\eta_2$ are the two elastic and two viscous parameters in the model, respectively and can be diagrammatically represented by the spring and dashpot schematic in Fig. 4b.

Solving the Burgers model for complex stiffness from Eqs. (4) and (6) gives,

$$\kappa^*(\omega) = \frac{(\tau_2 + \tau_3 + \tau_1^2\tau_3\omega^2)}{(1 - \tau_1\tau_3\omega^2)^2 + (E_1\omega)^2}\eta_2\omega^2 + i\frac{[1 + (\tau_1 + \tau_2)\tau_1\omega^2]}{(1 - \tau_1\tau_3\omega^2)^2 + (E_1\omega)^2}\eta_2\omega \tag{7}$$

where, $\tau_1 = \eta_1/\kappa_1$, $\tau_2 = \eta_2/\kappa_1$, $\tau_3 = \eta_2/\kappa_2$ and $E_1 = \tau_1 + \tau_2 + \tau_3$

Microrheology data from individual chromosomes were fit to the above equation by minimising the error function while solving a nonlinear least squares problem in MATLAB to extract $\kappa_1$, $\kappa_2$, $\eta_1$ and $\eta_2$ (Figs. 4 and S4).

### Indirect immunofluorescence and microscopy

Primary antibodies were used as follows: NPM1 (Mouse monoclonal, #32-5200 (Thermo fisher, Waltham, Massachusetts, USA) 1:60; Nucleolin (Rabbit polyclonal, Abcam, Cambridge, UK, #ab22758) 1:100. Fluorescence-labelled secondary antibodies; anti-mouse IgG-TRITC (Jackson ImmunoResearch, Ely, UK) and anti-Rabbit IgG-TRITC (Jackson ImmunoResearch, Ely, UK) were applied at dilution of 1:400. For immunofluorescence, cells were fixed in 3.5% paraformaldehyde for 15 min, permeabilized in 0.3% Triton X-100 for 5 min and blocked in 2% blocking solution (Thermo fisher). Cells were incubated overnight with the primary antibodies, washed in PBS, and secondary antibodies were applied for 1 h before counter-staining with DAPI. Fluorescent image acquisition was performed using a Leica (Wetzlar, Germany) TCS SPE confocal microscope with an x63 oil immersion objective. Microscope settings, including laser power, exposure time and pinhole size, were kept constant across all images in Fig. S1. Images were exported as TIFF files.

### Image analysis

Quantification of fluorescence intensity was performed in ImageJ version 1.54[73]. For chromosome images from single-chromosome experiments (Fig. 1b), fluorescence intensity was measured as the integrated density from line scans across each chromosome, divided by the length of the chromosome. NPM1 and nucleolin levels with Ki-67 manipulation were quantified from multichannel immunofluorescence microscopy images of early mitotic (prometaphase and metaphase) cells. Masks were generated by thresholding the DAPI channel to outline chromosomes and the NPM1 or nucleolin channel, respectively, to mark the spread of these proteins within the cell. The peri-chromosomal levels of NPM1 and nucleolin were measured as the mean intensity within the intersection of the two masks (using the AND function on the masks in the Image Calculator settings) in the respective image channel. Ki-67 levels were measured as the mean intensity within the chromosome mask in the mEmerald channel.

### Statistics and reproducibility

No statistical method was used to predetermine sample size. All attempts at replication were successful. Data were collected over several months, and two different instruments (at the University of Nottingham and the University of Kent) showed consistent results. Each experiment type had at least three independent replicates (from tissue culture to optical tweezer manipulation). Only chromosomes between 2 and 5 μm in length were analysed to ensure single chromosomes with telomeric attachment were being tested. Damaged and unravelled chromosomes were discarded on visual inspection. Data were also excluded if chromosomes were detached from the bead handles during the optical tweezer manipulations. Experiments were not randomised, but data collection methods were kept consistent, and scripts were used to automate analysis to remove bias.

Kruskal–Wallis tests with post-analysis multiple comparisons and linear regression were performed in MATLAB (2022b, MathWorks). $P$ values of ≤0.05 were considered to represent significance.

### Reporting summary

Further information on research design is available in the Nature Portfolio Reporting Summary linked to this article.

## Data availability

Data for the figures are provided with this paper as a 'Source Data' file. Raw data for microrheology analysis generated in this study, as well as Burgers model fits to data, can be accessed on Zenodo: https://doi.org/10.5281/zenodo.15632578. Code for this analysis is available on GitHub (see Code availability statement). Oscillatory microrheology data from Meijering et al.[40] plotted in Fig. 2e, f can be accessed on DataverseNL: https://doi.org/10.34894/XFZZPJ in the file: all_freq_dependent_stiffnesses.mat. Source data are provided with this paper.

## Code availability

The version of the i-Rheo C-Stretch app and associated code used to generate data in this paper can be accessed on Zenodo: https://doi.org/10.5281/zenodo.14527725.

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

## Acknowledgements

This work was funded by a Leverhulme Trust research project grant (RPG-2021-118) awarded to D.G.B. and A.J.W. Data acquisition at the University of Kent was enabled by a Travelling Fellowship to T.M. from the Company of Biologists (JCSTF23081207). The LUMICKS C-trap instrument at the University of Kent was funded by BBSRC (BB/T017767/1) to N.M.K., and one at the University of Nottingham was funded by a BBSRC Alert grant (BB/X019837/1) to D.G.B., T.M. and A.J.W. D.G.B. thanks: BBSRC David Phillips Fellowship (BB/V005626/1), Royal Society (RGS/R2/202366), Academy of Medical Sciences Springboard Award (SBF006\1071), and the Wellcome Trust (G.C., Drug Discovery and Team Science DTP 218466/Z/19/Z).

## Author contributions

The study was conceived by D.G.B., A.J.W., N.M.K and T.M., with funding secured by D.G.B. and A.J.W. N.M.K. provided access to a LUMICKS C-trap instrument at UoK. N.M.K. and R.U. provided training and assistance with data acquisition and analysis. The Ki-67-mEmerald cell line and Ki-67 overexpression plasmid were produced and characterised by K.L. T.M. carried out experimental work (chromosome isolation, chromosome manipulations and data analysis). Broadband 'stretch-hold' microrheology and the 'i-Rheo C-Stretch' app were developed by T.M. and M.T. M.T. derived the Burgers model equations for parameterising chromosome microrheology data. G.C. performed immunofluorescence staining and fluorescence confocal experiments. All authors contributed to data interpretation and the finished paper.

## Competing interests

The authors declare no competing interests.
