## [Transparent Peer Review file · Nature Communications]

The Mitotic Chromosome Periphery Modulates Chromosome Mechanics

Corresponding Author: Dr Daniel Booth

Version 0:

Reviewer comments:

Reviewer #1

(Remarks to the Author)

Manuscript: "The Mitotic Chromosome Periphery: A Fluid Coat That Mediates Chromosome Mechanics" by T. Mendonca et al

The manuscript describes optical trapping experiments on chromosomes isolated from a custom Ki-67-mEmerald HeLa cell line. The protein Ki-67 is addressed because of its role as "chief organizer of the mitotic chromosome periphery (MCP)". The authors study chromosomes from wild-type (WT), with over-expression (OE) and knock-down (KD) of Ki-67. The optical trapping experiment consists of a double optical trap (commercial instrument from LUMICKS), with microspheres bound to the two "ends" of the chromosomes.

With these chromosome-double-bead constructs (dumb-bell), the authors conduct "stretch-hold" experiments with the goal to explore both force-extension and microrheological properties of the chromosomes from the three different origins. For the force-extension analysis, the authors in particular extract the two material parameters Stretch modulus (S) and Stiffening Exponent (γ). In the viscoelastic analysis, the real and imaginary part of the stiffness (κ' and κ'') are extracted as function of angular frequency. These data are then fitted to the Kelvin representation of the Burgers model for a viscoelastic material, and associated parameters extracted.

In general, the manuscript is supplied with clear illustrations and conclusions are to some extent supported by the experimental data presented.

Concerns to be addressed are:

For the general discussion of (broadband) microrheology with optical traps, it is suggested that the authors also include discussion of active-passive broadband microrheology with optical tweezers as described in R. Kumar et al, *Sci. Rep* (2021)11:13917. Although the frequency range in that reference is substantially smaller than in the present manuscript, the method is claimed to work also inside viscoelastic materials.

The authors isolated chromosomes using a common protocol based on a PA buffer. However, since Ki-67 that is responsible for MCP organization was either knocked-down or over-expressed in the experiments, it raises the question as to whether the PA isolation protocol could have affected the MCP structure of these modified chromosomes. It would be beneficial if the authors clarified this concern, either through a discussion of the literature or by providing appropriate controls.

The "stretch-and-hold" broad-band method described in the manuscript appears to give consistent and smooth results for the elastic component, κ' , however always more seemingly noisy data for the viscous component, κ'' . Similar trends appear in other papers using conceptually similar methods from the Tassieri group. A discussion hereof would improve the paper and should be required for a high-profile publication like Nature Communications.

Figure 4 is confusing. First, the fits shown on top of many datasets recorded for the WT are plotted in a way that does not allow the reader to judge the quality of fits. Second, similar figures are not shown for the OE and KD chromosomes. As this is a major selling point for the paper, the authors should provide more evidence.

In conclusion, the authors describe an interesting and relevant study, that could be interesting for many of the readers of Nature Communications, however, the comments on the methodology should be addressed before publication can be

recommended.

(Remarks on code availability)

Reviewer #2

(Remarks to the Author)

In this work, the authors investigated the micromechanical properties of mitotic chromosomes. The authors used optical tweezers and a microrheology analysis to characterize the importance of the mitotic chromosome periphery (MCP) and changed the levels of Ki-67 by knocking out or expressing it. I have some comments and questions that could help to clarify some aspects of the text and the results:

Could the authors comment on how the stretching rates and piconewton forces applied in this study compare to the physiological rates and forces? It could be interesting to compare and expand the discussion on these values exerted by the mitotic spindle during chromosome segregation in anaphase. Also, it would be interesting to discuss the biological aspects of such forces and rates and what this implies for the MCP's role in participating in chromosome separation.

Could the authors expand on the discussion about the stiffening exponent decreases in KD chromosomes but not in OE chromosomes? Are there any structural aspects of MCP overexpression in a hierarchical organization, or does it exist only at the surface?

Could the authors expand on the different rates and how they may be associated with motor activity by SMC complexes or other molecular interactions?

(Remarks on code availability)

Reviewer #3

(Remarks to the Author)

The article by Mendonca et al. investigates the micromechanical properties of mitotic chromosomes, focusing on the contribution of the mitotic chromosome periphery (MCP). Using optical tweezers and a microrheology technique, the researchers provide a comprehensive characterization of single-chromosome mechanics across a broad frequency range. They show that the MCP may act as a key structural component that governs high-frequency self-reorganization dynamics and provides force-damping properties to mitigate mitotic stress. The study highlights the role of Ki-67 in organizing the MCP and its contribution to chromosome dynamics. This work can in principle advance our understanding of chromosome micromechanics and the fundamental properties of chromosomes. However, given that this work is a quantitative mechanical study, there are significant concerns about the quality of the work as discussed below.

1. One of the important mechanical parameters is the stretch modulus as shown in Figure 1. The authors stated under Methods that "The stretch modulus S was computed as the slope of force against normalised chromosome extension, $(L(t)-L_0)/L_0$, where $L(t)$ is the length of the chromosome at the given time and L_0 is its initial untangled length before any extension was registered as an increase in force." This definition ("registered as an increase in force") does not provide a clear definition of the force threshold for consideration. Because the force should asymptotically approach zero as the distance decreases, L_0 is ill-defined, creating uncertainties for data analysis. It is essential to clearly explain how L_0 and the stretch modulus are obtained using an example plot in Figure 1. In addition, a histogram or scatter plot of L_0 should be shown along with an explanation for the cause of the spread in L_0 .

2. It is unclear what the chromosome configuration is during stretching, and how this configuration is identified. Without this clarification, several configurations are possible: 1) each bead is attached to both sister chromatids; 2) each bead is attached to the same single chromatid; 3) each bead is attached to a different chromatid; 4) one bead is attached to one chromatid while the other is attached to both chromatids. There are no discussions of these configurations in the main text or methods. If the data are obtained from a single, double, or mixed chromatids, this uncertainty can create significant issues with data interpretation. Figure 1b and Figure 2b show cartoons that imply scenario 1). But what is the method to select this configuration from the other possibilities? Could this have caused the large spread in the measured parameter values in Figure 1d and 1e? In addition, since the sister chromatids are labelled with biotin throughout, each bead can attach to any part of the two sister chromatids, inevitably creating large variations in the bead attachment point relative to the chromosome. This needs to be clearly discussed in the main text.

3. The data in Figure 1 are intended to highlight the difference in mechanical properties of KD, WT, and OE chromosomes. However, only a single trace for the WT chromosome is shown at each stretch rate. Figure 1bi and 1bii should be expanded also to show individual force vs distance plots for KD and OE chromosomes for both the slow (0.02 $\mu\text{m/s}$) and fast (0.2 $\mu\text{m/s}$) stretching. For each trace, both the forward and reverse stretching should be shown to gain information on reversibility, hysteresis, and response time. In addition, histograms of L_0 for KD and OE chromosomes should also be added to Figure 1.

4. The authors argue for the importance of mechanical properties. However, the measured parameters in Figures 1d and 1e show a spread from many folds to an order of magnitude for each chromosome type but have a less than 2-fold difference among KD, WT, and OE chromosomes. These data do not bode well for the role of MCP in mechanical properties.

5. The Methods section mentions that the net stretching force was “the absolute sum of the force acting at both beads.” This contradicts Newton’s second law, which requires the forces of the chromosome on the two beads to be equal and opposite, so the force on the chromosome should be read from the force on a single bead. This is a rather gross error that impacts all measurements in this manuscript.

6. The abstract states, “We report that the MCP governs high-frequency self-reorganisation dynamics...” However, data in Figure 3c do not provide evidence to support this statement. In Figure 3c, the green data are labeled ‘Control’, which is unclear but presumably refers to the WT. It is very puzzling why the KD and OE chromosomes behave more similarly to each other than with the WT chromosomes. What is the physical reason for the similar mechanical response of knockdown (KD) chromosomes or overexpressed (OE) chromosomes?

7. Equation 3 shows an incorrect definition of strain.

8. Equation 6 shows a parameter σ without defining it.

(Remarks on code availability)

Reviewer #4

(Remarks to the Author)

(Remarks on code availability)

Reviewer #5

(Remarks to the Author)

Manuscript: “The Mitotic Chromosome Periphery: A Fluid Coat That Mediates Chromosome Mechanics” by T. Mendonca et al

The manuscript describes optical trapping experiments on chromosomes isolated from a custom Ki-67-mEmerald HeLa cell line. The protein Ki-67 is addressed because of its role as “chief organizer of the mitotic chromosome periphery (MCP)”. The authors study chromosomes from wild-type (WT), with over-expression (OE) and knock-down (KD) of Ki-67. The optical trapping experiment consists of a double optical trap (commercial instrument from LUMICKS), with microspheres bound to the two “ends” of the chromosomes.

With these chromosome-double-bead constructs (dumb-bell), the authors conduct “stretch-hold” experiments with the goal to explore both force-extension and microrheological properties of the chromosomes from the three different origins. For the force-extension analysis, the authors in particular extract the two material parameters Stretch modulus (S) and Stiffening Exponent (γ). In the viscoelastic analysis, the real and imaginary part of the stiffness (κ' and κ'') are extracted as function of angular frequency. These data are then fitted to the Kelvin representation of the Burgers model for a viscoelastic material, and associated parameters extracted.

In general, the manuscript is supplied with clear illustrations and conclusions are to some extent supported by the experimental data presented.

My concerns are:

For the general discussion of (broadband) microrheology with optical traps, I suggest that the authors also include discussion of active-passive broadband microrheology with optical tweezers as described in R. Kumar et al, *Sci. Rep* (2021)11:13917. Although the frequency range in that reference is substantially smaller than in the present manuscript, the method is claimed to work also inside viscoelastic materials.

The authors isolated chromosomes using a common protocol based on a PA buffer. However, since Ki-67 that is responsible for MCP organization was either knocked-down or over-expressed in the experiments, it raises the question as to whether the PA isolation protocol could have affected the MCP structure of these modified chromosomes. It would be beneficial if the authors clarified this concern, either through a discussion of the literature or by providing appropriate controls.

The “stretch-and-hold” broad-band method described in the manuscript appears to give consistent and smooth results for the elastic component, κ' , however always more seemingly noisy data for the viscous component, κ'' . Similar trends appear in other papers using conceptually similar methods from the Tassieri group. A discussion hereof would improve the paper and should be required for a high-profile publication like Nature Communications.

Figure 4 is confusing. First, the fits shown on top of many datasets recorded for the WT are plotted in a way that does not allow the reader to judge the quality of fits. Second, similar figures are not shown for the OE and KD chromosomes. As this is a major selling point for the paper, the authors should provide more evidence.

In conclusion, the authors describe an interesting and relevant study, that could be interesting for many of the readers of Nature Communications, however, the comments on the methodology should be addressed before publication can be recommended.

(Remarks on code availability)

Version 1:

Reviewer comments:

Reviewer #1

(Remarks to the Author)

We thank the authors for the clarifying comments and believe that our concerns have been well addressed. We therefore recommend publication of the manuscript in its present form.

(Remarks on code availability)

Reviewer #2

(Remarks to the Author)

The authors addressed all my questions and comments.

(Remarks on code availability)

Reviewer #3

(Remarks to the Author)

In this revised manuscript, the authors incorporated some of the suggested changes, which have improved the manuscript's clarity. However, they are reluctant to consider other suggestions for showing more data. If the authors do not want to show more data in the main figures, they should show the data as SI figures.

(Remarks on code availability)

Reviewer #5

(Remarks to the Author)

The Authors have addressed all the comments and made significant efforts to improve their manuscript. This work substantially advances our understanding of chromosome micromechanics and the role of the MCP in contributing to the fundamental properties of chromosomes. In its current form, it represents a valuable contribution to the field and is worthy of the readers' attention. The methodology is sound, and the results support the conclusions and claims.

Thank you for an engaging scientific discussion—it was a pleasure.

(Remarks on code availability)

n/a

Reviewer comments have been reproduced below in red text with our point-by-point responses in line. Page and line numbers mentioned refer to the marked-up document attached.

REVIEWER COMMENTS

Reviewer #1 (Remarks to the Author):

Manuscript: “The Mitotic Chromosome Periphery: A Fluid Coat That Mediates Chromosome Mechanics” by T. Mendonca et al

The manuscript describes optical trapping experiments on chromosomes isolated from a custom Ki-67-mEmerald HeLa cell line. The protein Ki-67 is addressed because of its role as “chief organizer of the mitotic chromosome periphery (MCP)”. The authors study chromosomes from wild-type (WT), with over-expression (OE) and knock-down (KD) of Ki-67. The optical trapping experiment consists of a double optical trap (commercial instrument from LUMICKS), with microspheres bound to the two “ends” of the chromosomes.

With these chromosome-double-bead constructs (dumb-bell), the authors conduct “stretch-hold” experiments with the goal to explore both force-extension and microrheological properties of the chromosomes from the three different origins. For the force-extension analysis, the authors in particular extract the two material parameters Stretch modulus (S) and Stiffening Exponent (γ). In the viscoelastic analysis, the real and imaginary part of the stiffness (κ' and κ'') are extracted as function of angular frequency. These data are then fitted to the Kelvin representation of the Burgers model for a viscoelastic material, and associated parameters extracted.

In general, the manuscript is supplied with clear illustrations and conclusions are to some extent supported by the experimental data presented.

Concerns to be addressed are:

For the general discussion of (broadband) microrheology with optical traps, it is suggested that the authors also include discussion of active-passive broadband microrheology with optical tweezers as described in R. Kumar et al, Sci. Rep (2021)11:13917. Although the frequency range in that reference is substantially smaller than in the present manuscript, the method is claimed to work also inside viscoelastic materials.

We would like to thank the Reviewer for their recommendation. We have now included a brief discussion on existing microrheology techniques. We have amended the text on page 3, line 184 as follows (new text highlighted in yellow): *‘While micromanipulation tools such as optical tweezers and magnetic tweezers have been used to study interphase chromosomes (Keizer 2022, Hameed 2012, Tseng 2004), their potential for broadband characterization of mitotic chromosome dynamics remains underexplored. In whole cells (Hardiman 2022) and tissue cultures (Mendonca 2023), optical tweezers have been used for passive broadband microrheology, where the optical trap is used purely for the spatial confinement of the bead probe. Active-passive approaches that combine optically driven perturbation of the bead probe with broadband passive measurements (Fischer 2010, Kumar 2021) have been used for microrheology of viscoelastic liquids. However, these active-passive approaches are multistep procedures that require sophisticated frequency and phase modulation of the active perturbation. In contrast, we introduce a bespoke single-step optical tweezer-based stretch-*

hold method, inspired by a similar approach used for microrheology of cells with atomic force microscopy (Chim 2018, Haidar 2024), to extract the mechanics of individual mitotic chromosomes across a broad frequency range.'

The authors isolated chromosomes using a common protocol based on a PA buffer. However, since Ki-67 that is responsible for MCP organization was either knocked-down or over-expressed in the experiments, it raises the question as to whether the PA isolation protocol could have affected the MCP structure of these modified chromosomes. It would be beneficial if the authors clarified this concern, either through a discussion of the literature or by providing appropriate controls.

Polyamines including spermine and spermidine in the PA buffer stabilize chromosomes while chelators (EDTA and EGTA) prevent nucleolytic digestion of the DNA. A number of pioneering studies done in the 70s and 80s have shown polyamine:EDTA containing buffers such as the PA buffer used in our study to be superior to other aqueous buffers at preventing cytoskeletal contamination (Lewis and Laemmli, *Cell* 1982), and generally conserving *in vivo* chromosome structure (Blumenthal et al., *J. Cell Biology* 1979) as evidenced by high resolution electron microscopy (Earnshaw and Laemmli, *J. Cell Biology* 1983). Crucially, chromosomal proteins (including those of the MCP – including several newly discovered ones) are still present after chromosome isolation with polyamine buffer – used in a study where ~4000 proteins were identified using mass spectrometry based proteomics (Ohta et al., *Cell* 2010). We also know that the MCP is robust, retaining its structure following harsh preparations such as chromosome spreads (Booth et al., *Mol. Cell* 2016) and DNase treatment (Matheson and Kaufman, *MBoC* 2017).

Recent research into chromosome micromechanics (Meijering et al., *Nature* 2022 and Witt et al., *Nat. Mater.* 2024) have employed this very same PA buffer. The Witt study describes chromosomes as polyelectrolyte gels that can change volume when the ionic strength of the buffer is altered. In this context, the Reviewer's suggestion of the buffer having an effect on the MCP structure may be tenable. However, from our observations we do not find OE (Ki-67 overexpression) chromosomes which are expected to possess an excess of charged domains, to appear larger or decondensed (see histogram of chromosome lengths below) so either, i) the ionic strength of the PA buffer is able to compensate for this increase in charged domains, or ii) the MCP may not significantly contribute to chromosome polyelectrolyte properties. We believe this warrants a separate new study.

	Mean ± SD	n
WT	3.6 ± 1.9 μm	104
OE	3.2 ± 1.7 μm	113
KD	3.5 ± 1.9 μm	94

Histogram of the initial natural lengths ' L_0 ' of all captured chromosomes with summary table. Only chromosomes between 2-5 μm in length were analysed to exclude centromere captures,

chromosome fragments and aggregates. We now include the histogram in **Supplementary Figure S3**

In this work, we used PA buffer with standard physiological concentrations of polyamines as a first step to determining the contribution of the MCP to chromosome micromechanics. While we believe that for the purpose of chromosome isolation, this buffer provides the best conditions for the preservation of the chromosome structure, we agree with the Reviewer that altering buffer conditions alongside MCP manipulations during the experimental procedure would be an interesting next research direction to begin to unravel the role of the MCP in chromosome volume phase transition and charge-mediated chromosome dispersal.

We now add a discussion of the above in line 394 on page 6: *‘Transient cross-links driven by charged interactions with intrinsically disordered proteins like Ki-67 may explain the dynamic, ‘fluid’ nature of the MCP. Such charged interactions have recently been shown to influence chromosome fluidity and can be modified by altering the ionic strength of the experimental buffer (Witt 2024). Chromosomes behave as polyelectrolyte gels where charged interactions maintain osmotic pressure, as described by Donnan’s theory (Witt 2024, Beel 2021), to support chromosome structure. These polyelectrolyte properties may be driven by the MCP alongside histones. Our data however, show OE chromosomes which are expected to possess an excess of charged domains, do not appear decondensed (Figure S3d). It is therefore probable that, i) the experimental polyamine (PA) buffer (see Methods) is able to compensate for this increase in charged domains, or ii) the MCP may not significantly contribute to chromosome polyelectrolyte properties and therefore, chromosome condensation.’*

We also amend the Methods section to include on line 492, page 7: *‘Growth media was supplemented with biotin (50 mM) 24 hours before chromosome isolation using a protocol similar to previously described (Meijering 2022) using polyamine (PA) buffer to enhance purity of isolated chromosomes and retain chromosome structure (Earnshaw 1983, Lewis 1982, Ohta 2010).’*

The “stretch-and-hold” broad-band method described in the manuscript appears to give consistent and smooth results for the elastic component, κ' , however always more seemingly noisy data for the viscous component, κ'' . Similar trends appear in other papers using conceptually similar methods from the Tassieri group. A discussion hereof would improve the paper and should be required for a high-profile publication like Nature Communications.

We appreciate the Reviewer’s familiarity with Tassieri’s work, and we are pleased to address this concern.

In ideal (i.e., noise-free) systems, the analytical approach developed by Tassieri et al. (i.e., i-Rheo) operates without issues, providing accurate viscoelastic moduli. However, when applied to real experimental data, the output of the analytical approach is inevitably influenced by noise present in the input measurements. **This effect is most pronounced in the less dominant of the two viscoelastic components as with the same level of noise, the signal to noise ratio here will be lower.**

In the case of chromosomes—given their soft-solid nature—the elastic modulus dominates over most of the explored frequency range, particularly at high frequencies, where its signal

remains higher and thus less affected by measurement noise. Conversely, the viscous component, being inherently smaller, is more susceptible to noise-induced fluctuations.

This is illustrated in the image below, which shows the viscoelastic moduli derived from an ideal single-mode Maxwell model, characterized by a single exponential decay (first case, purely analytical). The second case in the figure displays the viscoelastic moduli of the same single exponential decay, analysed using i-Rheo GT—a module of Tassieri’s i-Rheo analytical tools—which implements the same underlying Fourier transform method (i-Rheo FT) used in our complex stiffness $\kappa^*(\omega)$ measurements. The third case in the example includes added noise, highlighting how noise primarily affects the viscous modulus G'' at high frequencies, where the elastic modulus G' dominates. Conversely, at low frequencies, where G'' is dominant, the noise impacting G' becomes more apparent. Reviewers are invited to verify this explanation using i-Rheo-Web, which is freely available online: <https://i-rheo.mib3avenger.com/>

$G(t)$ is the material’s shear relaxation modulus, $G^*(\omega)$ is the complex modulus and $\hat{G}(\omega)$ is the Fourier transform of $G(t)$. Γ is the material characteristic relaxation frequency, t is time and ω is frequency.

We now include a discussion in the Methods section of the revised manuscript on page 9 line 641 to clarify: ' $\kappa^*(\omega)$ and $\tan\delta$ were acquired using a MATLAB application i-Rheo C-Stretch (see Code availability and Figure S5) that implements a Fourier transform method introduced previously (Smith 2021, Tassieri 2016, Haidar 2024). Like any numerical method, this analytical approach is also influenced by the level of noise present in the input measurements. The impact of noise is most pronounced on the smaller of the two viscoelastic components, as the signal to noise ratio here will be lower. In this study, the elastic modulus dominates across most of the explored frequency range, particularly at high frequencies, making it less susceptible to measurement noise. In contrast, the viscous component, being predominantly smaller, is more sensitive to noise-induced fluctuations.'

Figure 4 is confusing. First, the fits shown on top of many datasets recorded for the WT are plotted in a way that does not allow the reader to judge the quality of fits. Second, similar figures are not shown for the OE and KD chromosomes. As this is a major selling point for the paper, the authors should provide more evidence.

We have updated Figure 4 to remove the previous panel a, which showed the model fit to WT data and a new Supplementary Figure S4 has been added to better show Burger model fits to the microrheology data for all three conditions (WT, KD and OE) in response to the Reviewer's recommendation. We have also amended the description of the model fitting in the Methods line 666 as follows: '*Microrheology data from individual chromosomes were fit to the above equation by minimising the error function while solving a nonlinear least squares problem in MATLAB to extract κ_1 , κ_2 , η_1 and η_2 (Figures 4, S4).*'

In conclusion, the authors describe an interesting and relevant study, that could be interesting for many of the readers of Nature Communications, however, the comments on the methodology should be addressed before publication can be recommended. We are pleased that the Reviewer recognises the merits of our study and thank them for their positive and favourable recommendation. We agree that this work will be of interest to a broad range of readers of Nature Communications across the disciplines of cell biology, cancer, biophysics, optics and material sciences to name a few.

Reviewer #2 (Remarks to the Author):

In this work, the authors investigated the micromechanical properties of mitotic chromosomes. The authors used optical tweezers and a microrheology analysis to characterize the importance of the mitotic chromosome periphery (MCP) and changed the levels of Ki-67 by knocking out or expressing it. I have some comments and questions that could help to clarify some aspects of the text and the results:

Could the authors comment on how the stretching rates and picoNewton forces applied in this study compare to the physiological rates and forces? It could be interesting to compare and expand the discussion on these values exerted by the mitotic spindle during chromosome segregation in anaphase. Also, it would be interesting to discuss the biological aspects of such forces and rates and what this implies for the MCP's role in participating in chromosome separation.

We are grateful to the Reviewer for their insightful questions which have helped us add valuable biological context to our manuscript.

The following text has now been included on page 2:

Line 98: 'This picoNewton sensitivity makes optical tweezers ideal for testing chromosome mechanics at biologically relevant forces. Studies across species and cell types have reported forces ranging from tens to hundreds of picoNewtons acting on individual chromosomes throughout mitotic progression; from congression to bio-orientation and segregation (Nicklas 1988, Ye 2016, Anjur-Dietrich 2021). During anaphase A, pulling forces of 200-700 pN have been recorded at chromosomes to move them towards spindle poles in grasshopper spermatocytes (Nicklas 1983). The chromosome manipulations in this study were within this range with a maximum stretching force of ≈ 400 pN.'

Line 114: 'At a force loading-rate of $0.02 \mu\text{m/s}$ which closely corresponds to the recorded speed for oscillations of human chromosomes at the metaphase plate (Amaro 2010), and the rate of directed movement of chromosomes during anaphase in mammalian cells (Chong 2024) and grasshopper spermatocytes (Nicklas 1965), a linear increase in force is required to stretch the chromosome greater distances (Man 2021).'

Line 132: 'At a faster stretching rate of $0.2 \mu\text{m/s}$ relevant to rates at which microtubule depolymerisation-led chromosome motion was observed in vitro (Coue 1991), chromosome moving chromo-kinesin motor proteins function in mammalian cells (Brouhard 2005) and SMC motors have been recorded extruding DNA loops in yeast (Ganji 2018), the force-extension relationship bears a non-linear form in a majority of tested chromosomes (Figure 1bii and c)'

Page 3:

Line 216: 'Although chromosomes would not be deformed at $100 \mu\text{m/s}$ in nature, these fast manipulations allow us to probe molecular scale interactions along with macro-scale mechanical properties to understand the contribution of the MCP. This fast stretching was followed by a dwell period of 2 minutes, during which the positions of both optical traps were held constant.'

Could the authors expand on the discussion about the stiffening exponent decreases in KD chromosomes but not in OE chromosomes? Are there any structural aspects of MCP overexpression in a hierarchical organization, or does it exist only at the surface? The Reviewer's comment is pertinent and has helped us add relevant background to the discussion of these results. Structural organisation of the MCP is underexplored and while theoretical models have been proposed, conclusive experimental data is required. Current thinking (although this is yet to be confirmed at the ultra-structural level) is that Ki-67 extends from the chromosome surface in a brush-like organisation (Cuylet et al, *Nature* 2016), but there is less agreement on how other MCP components associate with this Ki-67 scaffold. One possibility is a random colocalisation of these proteins and RNA or alternatively, a layered organisation with RNA forming linkages between Ki-67 and outer MCP proteins (Hayashi et al., *Biochem. Biophys. Res. Commun* 2017, Ma et al., *Cell Discov.* 2022). This is an area of the MCP that we are actively working on.

We have now added a discussion to our manuscript to expand on the stiffening exponent results in line 161: 'The structural organisation of the MCP remains underexplored. Current speculation places Ki-67 in extended polarised brush conformation at the chromosome surface, acting as a scaffold for other MCP proteins and RNA (Cuylet 2016, Hayashi 2017, Ma 2022). This structural model predicates an increase in spatial density of the MCP in OE chromosomes, which may result in a disordered MCP cross-linked network, explaining the broad variation in our results.'

We also now add interpretation to our Stretch Modulus results to say that MCP interaction is unlikely to be with the chromosome scaffold in line 126: 'S remains unchanged with varying

MCP levels (Figure 1d). Since S is directly related to the bending stiffness and persistence length of flexible chain polymers which chromosomes can be modelled as (Broedersz 2014), our results suggest that the MCP does not interact with the chromosome scaffold, its backbone.'

Could the authors expand on the different rates and how they may be associated with motor activity by SMC complexes or other molecular interactions?

We have now added some context around SMC motor protein rates in line 132 (see response to question 1 above) and discuss SMC complexes and molecular interactions in our manuscript (see line 325 onwards on page 5). As the SMC complexes were not the focus of our study and were not subjected to experimental manipulations, we have not expanded further on this. We do now say that our rheology method would enable future investigations of these interactions in line 335.

Reviewer #3 (Remarks to the Author):

The article by Mendonca et al. investigates the micromechanical properties of mitotic chromosomes, focusing on the contribution of the mitotic chromosome periphery (MCP). Using optical tweezers and a microrheology technique, the researchers provide a comprehensive characterization of single-chromosome mechanics across a broad frequency range. They show that the MCP may act as a key structural component that governs high-frequency self-reorganization dynamics and provides force-damping properties to mitigate mitotic stress. The study highlights the role of Ki-67 in organizing the MCP and its contribution to chromosome dynamics. This work can in principle advance our understanding of chromosome micromechanics and the fundamental properties of chromosomes. However, given that this work is a quantitative mechanical study, there are significant concerns about the quality of the work as discussed below.

1. One of the important mechanical parameters is the stretch modulus as shown in Figure 1. The authors stated under Methods that "The stretch modulus S was computed as the slope of force against normalised chromosome extension, $(L(t)-L_0)/L_0$, where $L(t)$ is the length of the chromosome at the given time and L_0 is its initial untangled length before any extension was registered as an increase in force." This definition ("registered as an increase in force") does not provide a clear definition of the force threshold for consideration. Because the force should asymptotically approach zero as the distance decreases, L_0 is ill-defined, creating uncertainties for data analysis. It is essential to clearly explain how L_0 and the stretch modulus are obtained using an example plot in Figure 1. In addition, a histogram or scatter plot of L_0 should be shown along with an explanation for the cause of the spread in L_0 .

We thank the Reviewer for their valuable feedback. We are grateful that this peer review process has enabled us to make corrections and improve our manuscript.

We have clarified the definition of L_0 on page 8:

Line 544: 'Experimental manipulations were performed using this dumbbell configuration with the optical traps aligned along the X-axis of the C-trap imaging system. At the start of every experiment, before any stretching forces were applied to the chromosome, force at both beads in the chromosome dumbbell were zeroed to set a baseline force reference. One optical trap was then stepped a small distance to ensure attachment of the chromosome to both beads

and to bring it to its natural length L_0 , defined as the distance between the bead handles before any further stepping of a bead resulted in resistive force from the chromosome, above the set baseline.'

588: 'The stretch modulus S was computed as the slope of force against normalised chromosome extension, $(L(t)-L_0)/L_0$, where $L(t)$ is the length of the chromosome at the given time and L_0 is the natural length of the chromosome.'

The histogram below of L_0 has now been included in supplementary Figure S3.

2. It is unclear what the chromosome configuration is during stretching, and how this configuration is identified. Without this clarification, several configurations are possible: 1) each bead is attached to both sister chromatids; 2) each bead is attached to the same single chromatid; 3) each bead is attached to a different chromatid; 4) one bead is attached to one chromatid while the other is attached to both chromatids. There are no discussions of these configurations in the main text or methods. If the data are obtained from a single, double, or mixed chromatids, this uncertainty can create significant issues with data interpretation. Figure 1b and Figure 2b show cartoons that imply scenario 1). But what is the method to select this configuration from the other possibilities? Could this have caused the large spread in the measured parameter values in Figure 1d and 1e? In addition, since the sister chromatids are labelled with biotin throughout, each bead can attach to any part of the two sister chromatids, inevitably creating large variations in the bead attachment point relative to the chromosome. This needs to be clearly discussed in the main text.

Chromosomes used in our study were isolated from cells arrested in prometaphase using Nocodazole. At this stage, sister chromatids are bound together by cohesins and appear thread-like (Figure 1ai). These isolated chromosomes orient lengthwise along the fluid flow in the microfluidics channels, ensuring that chromosome capture is biased along the telomeres as in configuration 1 (from the Reviewer's stated cases). The optical traps are aligned horizontally to the X-axis of the imaging system of the Lumicks C-trap at the start of each measurement. If initial contact between the chromosome and beads is in either of the configurations 2-4, when the optical traps are moved apart, one or both beads rotate within their respective optical trap to bring the experimental dumbbell unit into the desired configuration 1 with the chromosome aligned along the X-axis of the dumbbell and the imaging system before any deformation of the chromosome can occur (see schematic below). This self-correction also ensures contact at each bead with both sister chromatids as they are conjoined. Rarely (<20 %, see histogram in response above) centromeric or 'hugged'

conformations as described in Meijering et al. (*Nature* 2022) occur where beads are attached to separate sister chromatids with the chromosome vertically in between. To avoid analysing these, we exclude cases where beads are $< 2\mu\text{m}$ apart as chromosomes have been shown to be $\sim 1.2\ \mu\text{m}$ wide and the shortest chromosomes are $\sim 2\ \mu\text{m}$ in length (Booth et al., *Mol. Cell* 2016). We also discard any chromosomes that look unravelled or clumped at the time of measurement by visual inspection. Measurements of chromosomes longer than $5\ \mu\text{m}$ are also excluded to avoid chromosome aggregates.

*Schematic of capture configurations; a. ideal telomeric capture of two sister chromatids at first bead or b. an example showing initial attachment at one chromatid. c1-4. Possible dumbbell conformations with 1. ideal attachment of both sister chromatids at both beads 2. only one but different chromatid at each bead 3. only one but the same chromatid at both beads and 4. one chromatid at one bead and two at the second bead. One bead was stepped a short distance which rotated one or both beads (dashed curved line) in cases 2-4 such that the long axis of the chromosome (black dashed line) co-aligns with the X-axis of the dumbbell (orange dashed line) to bring the dumbbell into the desired position in case 1. d. Final dumbbell conformation before further experimental manipulation. A modification of this schematic has now been included in the manuscript as a new **Supplementary Figure S3**.*

We have now added an explanation of this in the Methods section, page 8 line 547 following on from the text reproduced in the response above: 'At the start of every experiment, before any stretching forces were applied to the chromosome, force at both beads in the chromosome dumbbell were zeroed to set a baseline force reference. One optical trap was then stepped a small distance to ensure attachment of the chromosome to both beads and to bring it to its natural length L_0 , defined as the distance between the bead handles before any further stepping of a bead resulted in resistive force from the chromosome, above the set baseline. **This stepping rotated one or both beads within their respective optical trap in most cases to bring the chromosome to L_0 and in alignment with the X-axis of the imaging system and the dumbbell, before any deformation of the chromosome could occur (Figure S3).** Further

manipulations were performed in a third microfluidic channel containing only PA buffer as detailed in the sections below.'

We discuss the length selection criteria in line 566: 'Only chromosomes between 2-5 μm in length were analysed to ensure single chromosomes with telomeric attachment were being tested (Figure S3d). **Damaged and unravelled chromosomes were discarded on visual inspection.'**

3. The data in Figure 1 are intended to highlight the difference in mechanical properties of KD, WT, and OE chromosomes. However, only a single trace for the WT chromosome is shown at each stretch rate. Figure 1bi and 1bii should be expanded also to show individual force vs distance plots for KD and OE chromosomes for both the slow (0.02 $\mu\text{m}/\text{s}$) and fast (0.2 $\mu\text{m}/\text{s}$) stretching. For each trace, both the forward and reverse stretching should be shown to gain information on reversibility, hysteresis, and response time. In addition, histograms of L0 for KD and OE chromosomes should also be added to Figure 1.

Figure 1 is intended to highlight the rate dependence of chromosome mechanical properties, so depending on the speed of pulling, chromosomes respond differently – either showing a linear elastic response or a non-linear sequential stiffening response in a majority of cases (Figure 1c). These different responses are reveal different aspects of chromosome dynamics, from their low frequency (long time) elastic properties to their network microstructure. **This is highlighted by the title of the Figure** and we explain further in the text on page 2 line 110 onwards that these single frequency experiments in isolation present limited snapshots of chromosome complex mechanics. Figure 1 bi and bii are meant to simply illustrate this difference in response (irrespective of MCP status) and we have now amended the figure legend to clarify this.

We thank the Reviewer for their suggestions of the additional hysteresis and reversibility plots but these suggested experiments suffer from being single frequency in their nature and therefore again provide limited information about the chromosome's viscoelastic properties, which are time-dependent. Our microrheology method allows us to extract the viscoelastic properties of chromosomes across a broad frequency range using a simple, single-step method as opposed to performing several stretching experiments with different rates and extents. We also reference works by Poirier et al. (2000, 2001) that have reported such hysteresis measurements in chromosomes and place our results in context with these previous reports (line 110 and 172).

We now highlight in line 181: '*This prompted us to develop a novel broadband microrheology approach for single chromosomes in a single step as opposed to performing a multitude of stretching experiments with different rates and extents.'*

4. The authors argue for the importance of mechanical properties. However, the measured parameters in Figures 1d and 1e show a spread from many folds to an order of magnitude for each chromosome type but have a less than 2-fold difference among KD, WT, and OE chromosomes. These data do not bode well for the role of MCP in mechanical properties.

We would like to bring to the Reviewer's attention to the fact that we clearly state that under the slow stretch rate conditions in Figure 1d, there is no difference in the measured stretch modulus between the three conditions (see line 126). However, there is a clear statistical difference in non-linear sequential stiffening (Figure 1e) between KD and WT chromosomes. We report the statistics for this test on page 3 along with a discussion of our interpretation of

the results. The broad variation in data is typical for complex biomaterials like chromosomes. We go on to explain on page 3 that chromosomes have complex time-dependent properties that are not fully captured by the single frequency experiments in Figure 1 and introduce our broadband microrheology method for this purpose. Our microrheology data show clear differences between WT, KD and OE chromosomes (Figures 3 and 4) seen dominantly at high-frequencies which we attribute to differences in self-reorganisation with varying levels of the MCP. This highlights the importance of our new method for assessing the broadband response of chromosomes to stretching, and also for considering the appropriate rate at which chromosomes are stretched to measure specific dynamics.

5. The Methods section mentions that the net stretching force was “the absolute sum of the force acting at both beads.” This contradicts Newton’s second law, which requires the forces of the chromosome on the two beads to be equal and opposite, so the force on the chromosome should be read from the force on a single bead. This is a rather gross error that impacts all measurements in this manuscript.

We appreciate the Reviewer’s careful assessment of our methodology and acknowledge that this needs correcting.

We recognize that, according to **Newton’s third law**, the force exerted by the displaced bead on the chromosome is transmitted along the chromosome to the stationary bead, which reacts with equal magnitude and opposite direction to hold the chromosome in place. As a result, the actual force acting on the chromosome should be determined from the force measured on a single bead, rather than from the absolute sum of the forces on both beads.

We include here a box plot showing the distribution of forces acting on the both beads; one kept stationary and the other displaced for the microrheology measurement. As correctly anticipated by the Reviewer, the two forces are essentially equal in magnitude, within the bounds of experimental uncertainty. Given that the forces on the two beads and the force acting on the chromosome is the force measured from one bead, not the sum of the two beads, the actual force acting on the chromosome is half of what we originally considered. Importantly, the overall dynamics in terms of the **frequency dependence of the viscoelastic moduli remain unchanged**, apart from their magnitude, which is simply scaled by a factor of two; therefore, the conclusions of the study are still valid.

The box plot shows the absolute values of forces acting on each bead throughout the experiment (data from 5 experiments shown), where Bead 1 refers to the stationary bead and Bead 2 refers to the bead that is displaced during the experiment.

We also take this opportunity to clarify that the force on the displaced bead pulling the chromosome is defined as:

$$F_{\text{chromosome}} = F_{\text{optical}} - F_{\text{drag}}$$

where F_{drag} represents the frictional force exerted on the bead by the buffer. This drag force reaches its maximum (~3 pN) when the bead moves at ~100 $\mu\text{m/s}$ during the initial stretching step. However, since **our analysis focuses exclusively on the relaxation process**, during which the bead moves at only a few microns per second, we neglect this term. At such low speeds, the drag force is at least **two orders of magnitude smaller** than the force acting on the chromosome, making its contribution negligible.

We appreciate the Reviewer's feedback and have updated the manuscript accordingly in line 623: *'The force exerted by the displaced bead on the chromosome is transmitted along the chromosome to the stationary bead, which reacts with equal magnitude and opposite direction to hold the chromosome in place. The stretching force $F(t)$ acting on the chromosome was therefore calculated as the average of the force measured at both bead handles for accuracy.'*

6. The abstract states, "We report that the MCP governs high-frequency self-reorganisation dynamics..." However, data in Figure 3c do not provide evidence to support this statement. In Figure 3c, the green data are labeled 'Control', which is unclear but presumably refers to the WT. It is very puzzling why the KD and OE chromosomes behave more similarly to each other than with the WT chromosomes. What is the physical reason for the similar mechanical response of knockdown (KD) chromosomes or overexpressed (OE) chromosomes?

As the Reviewer identified, the OE results are unexpected. We interpret the discrepancy with the OE chromosomes to point to chromosomes requiring an optimal concentration of the MCP to maintain normal dynamics (see line 307: *'Furthermore, reorganisation appears to be suppressed in OE chromosomes, suggesting that an optimal MCP load is required for normal mechanical behaviour.'*). We have added some context to MCP structural organisation in line 161 in response to Reviewer #2's question which gives important context to this discussion. The current brush model of Ki-67 places it in an extended conformation on the chromosome surface, acting as a scaffold for other MCP proteins. This structural model allows us to assume that an enriched MCP is also more densely packed. We have added this context to the discussion on page 4 line 304 – page 5 line 319, where we postulate that molecular interactions are disordered in the over-crowded OE MCP resulting in OE chromosomes switching between WT and KD-like relaxation dynamics.

We also thank the Reviewer for bringing to our attention the oversight in the legend of Fig 3c. We have now corrected this.

7. Equation 3 shows an incorrect definition of strain.

The Reviewer is correct here, strain is the relative dimensionless change in chromosome length but because we consider the change in chromosome length in nanometres, we now refer to this variable as deformation ' λ ' and have now corrected this in equation 3 and throughout the manuscript.

8. Equation 6 shows a parameter sigma without defining it.

We thank the Reviewer for bringing this error to our attention. Sigma here represents the applied force. We now replace sigma with 'F' in equation 6 and defined it as such.

Reviewer #4 (Remarks to the Author):
